# Congenital pulmonic and aortic stenosis in Newfoundland dogs: Results of a 14-year French cardiovascular screening program (921 dogs)

**Valérie Chetboul**[1,2]*, **Constance Fauveau**[3], **Peggy Passavin**[4]

**1** Cardiology Department, École Nationale Vétérinaire d'Alfort, CHUVA, Maisons-Alfort, France, **2** INSERM, IMRB, Univ Paris Est Créteil, Créteil, France, **3** Surgical Department, École Nationale Vétérinaire d'Alfort, CHUVA, Maisons-Alfort, France, **4** Cardiology Department, Vétérinaire Clinic Boulogne Roland Garros, Boulogne Billancourt, France

\* valerie.chetboul@vet-alfort.fr

## Abstract

**Data Availability Statement:** All relevant data are within the paper and its Supporting Information files.

### Introduction

Aortic stenosis (AS) and pulmonic stenosis (PS) are two of the most common canine congenital heart diseases (CHD), with a high relative risk for Newfoundland dogs to develop inherited subvalvular AS. For this reason, a cardiovascular screening program has been set up by the French Newfoundland kennel club in order to manage mattings and reduce AS prevalence.

### Materials and methods

The records of untreated and non-anesthetized adult Newfoundland dogs screened between 2010 and 2023 were retrospectively reviewed. All dogs underwent physical examination and standard transthoracic echocardiography with concomitant ECG tracing. All examinations were reviewed by one single board-certified specialist in cardiology.

### Results

A total of 921 dogs were screened during the study period (female:male sex ratio = 1.94, median age [IQR] = 1.9 years [1.6–2.7], body weight = 55.0 kg [50–60]). For most dogs (90.6% for AS and 91% PS), a single examination was required to obtain a definitive cardiac status, although most operators (122/133 = 91.7%) were non-specialist general practitioners. Out of the 921 screened dogs, 913/921 (99.1%) and 919/921 (99.8%) were respectively free of AS and PS, with no AS and PS detection during the last 3 years of the program. The inbreeding coefficient, which was assessed from the pedigree analysis of all screened dogs except one, was not significantly different between dogs with either AS (0.59%; P = 0.86) or PS (0.39%; P = 0.72) and those without any arterial stenosis (0.39%).

**Funding:** The author(s) received no specific funding for this work.

**Competing interests:** The authors have declared that no competing interests exist.

## Conclusion

This 14-year cardiovascular screening program has experienced a strong involvement of veterinarians, breeders, and owners throughout France. Unlike reports from other European and North American countries, this program suggests the low and decreasing prevalence of both AS and PS in the Newfoundland breed in France.

## Introduction

Aortic stenosis (AS) and pulmonic stenosis (PS) are two of the most common congenital heart diseases (CHD) in dogs both consisting in respectively left and right ventricular outflow tract obstructions (VOTO) [1–7], with a high relative risk for Newfoundland dogs to develop sub-valvular AS (SAS), odds ratios reaching up to 35 as compared with other canine breeds [1, 2, 8–13]. In both AS and PS cases, obstruction to the ejection of blood from respectively the left and right ventricles leads to turbulences associated with increased of blood velocity across the stenotic lesion depending on the obstruction severity, and thus to the presence of a left basilar systolic heart murmur detected on cardiac auscultation. Subvalvular AS is characterized by various types of lesions, including fibrous or fibromuscular nodules, ridges, rings, and complex tunnel-like tissues located below the aortic valve, thus creating discrete to severe obstruction to blood flow across the left ventricular outflow tract [8, 13–15]. Dogs with moderate to severe SAS have been demonstrated to be at higher risk for developing clinical signs (e.g., exercise intolerance, syncope), sudden death, infective endocarditis, and left-sided congestive heart failure, with a median survival of 3.0 years only for dogs with Doppler-derived peak systolic trans-stenotic pressure gradient ($\Delta$P) > 130 mmHg [14, 15].

Subvalvular AS mainly affects large-sized dogs, with Newfoundland dogs being one of the most commonly overrepresented breeds [1, 2, 9, 13, 14]. Based on pathological examinations and breeding analysis, SAS was early suspected to be inherited in Newfoundland dogs [8]. The extensive pedigree analysis of more than 230,000 Newfoundland dogs from the European and North American population reaching back to the 19th century confirmed that SAS-affected Newfoundland dogs were more inbred than unaffected dogs, and an autosomal codominant mode of inheritance (with lethal homozygosity and a penetrance of 1/3 in the heterozygotes) was suggested [10]. Additionally, a single codon insertion in the Phosphatidylinositol Binding Clathrin Assembly Protein (*PICALM*) gene was shown to be associated with the development of SAS in a family of North American Newfoundland dogs [11], although this single variant was not identified in European Newfoundland dogs [16]. Therefore, the genetic basis of SAS in Newfoundland dogs still remains unresolved and further investigations are needed to identify causative mutations.

Given the high predisposition of Newfoundland dogs to SAS, the global high prevalence of PS in dogs, the absence of genetic tests for both CHD together with their potential severity, a protocol for screening Newfoundland dogs for both arterial stenoses was designed in France in 2009 and proposed to the French Newfoundland kennel club (called "*Club Français du Chien Terre Neuve et du Landseer*", Landseer dogs specifically designating black and white Newfoundland dogs in Europe). One year later, this protocol was approved and officially implemented under the supervision of one single board-certified specialist, i.e., Diplomate of the European College of Veterinary Internal Medicine (Cardiology, VC).

The objective of the present retrospective study was therefore to analyze the results of this 14-year Newfoundland cardiovascular screening program (2010–2023).

## Materials and methods

### Animals

The case records of 921 breeder- and client-owned Newfoundland dogs that underwent the French mandatory cardiovascular screening program for detecting AS and PS between January 2010 and December 2023 were retrospectively reviewed. No ethical committee was required for this study owing to its retrospective epidemiological aspect and because it does not refer to an experimental protocol used for teaching or research, but rather to a noninvasive clinical screening program for the French Newfoundland kennel club. Newfoundland dogs included in the program had to be at least 12 months old, free of medication, and non-anesthetized. For each dog, pedigrees and implanted microchips were checked and registered by the veterinarians involved in the program (doctors in veterinary medicine (DVM), either board-certified specialists or general practitioners), and both were then checked again by the Newfoundland kennel club committee.

### Protocol

The protocol was strictly similar to that of the national mandatory cardiovascular breed screening program set up in France for the Boxer breed [17]. The two screening programs (the present one on Newfoundland dogs and the previous one on Boxer dogs) were designed by the same board-certified cardiologist (VC) who was also the single official reviewer of Doppler examinations for both breeds, as requested by the national French kennel club [17]. Briefly, all Newfoundland dogs underwent a complete physical examination followed by standard transthoracic echocardiography (M-mode, two-dimensional (2D) mode, and spectral Doppler) with concomitant electrocardiographic (ECG) tracing. For each dog, all operators had to fill out an official form including epidemiological (animal's name, age, sex, body weight), clinical, echocardiographic, and Doppler data. If present, heart murmurs had to be described and categorized using a 6-level classification scheme [18]. Echocardiographic examinations could be performed with dogs in lateral recumbency or in standing position. Peak systolic aortic and pulmonary flow velocities were both measured using continuous-wave Doppler mode from respectively the left apical 5-chamber view or the subcostal view, and from the right or left short-axis view at the level of the aortic valve [19]. Good quality printed images had to be provided, including at least two well defined Doppler curves with concomitant ECG tracing. The corresponding 2D image had to clearly show the Doppler cursor position (parallel to vessel axis in center stream of the systolic arterial flow). The diagnosis of AS and PS was based on the following criteria (Fig 1): 1) systolic ejection heart murmur loudest in the left basilar area, and 2) peak aortic flow velocity > 2.5 m/s for AS and peak pulmonary flow velocity > 2.0 m/s for PS [20]. The modified Bernoulli equation was applied to peak systolic flow velocities to calculate the corresponding $\Delta P$ (in mmHg). Both AS and PS severities were defined as follows: mild for $\Delta P \leq 50$ mmHg, moderate for $\Delta P > 50$ and $\leq 80$ mmHg, and severe for $\Delta P > 80$ mmHg [17, 20].

In order to ensure data quality, all forms and Doppler examinations were reviewed and validated by one single board-certified specialist, i.e., Diplomate of the European College of Veterinary Internal Medicine (Cardiology, VC). The board-certified official reviewer rejected Doppler examinations if Doppler quality criteria were not fulfilled, e.g., 2D images of poor quality, unclearly defined Doppler curves, incorrect placement of the Doppler cursor, or absence of concomitant ECG tracing. In such cases, Doppler examinations were asked to be redone. For equivocal cases, additional information (e.g., pulsed-wave and color-flow Doppler examinations, and 2D images showing evidence of obstructive lesions) were required by the official reviewer before a definitive diagnosis could be provided.

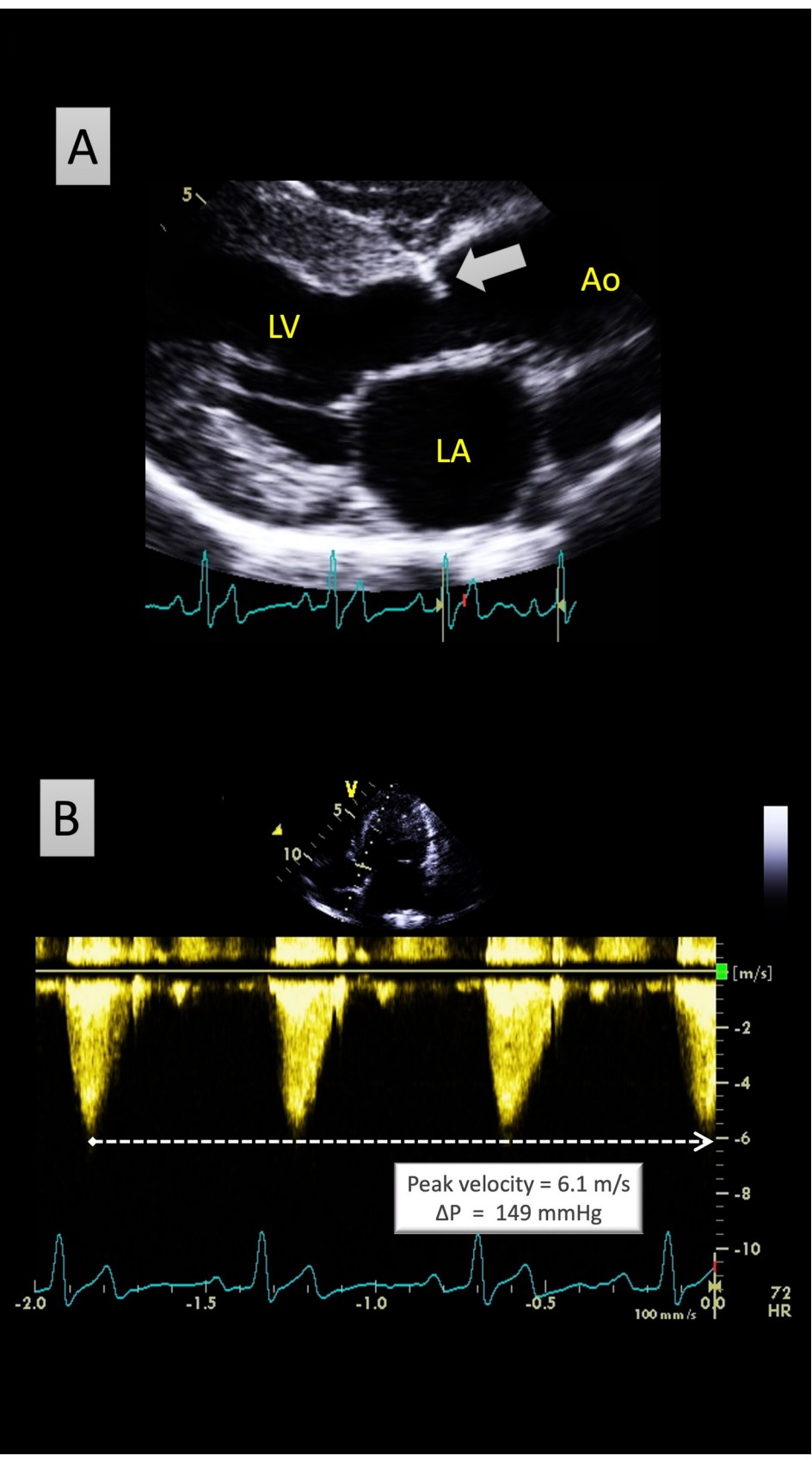

**Fig 1.** Representative two-dimensional (A) and continuous-wave Doppler echocardiographic view (B) from a Newfoundland dog with severe congenital subaortic stenosis. The two-dimensional right parasternal 5-chamber view shows a hyperechoic fibrous ridge (arrow) within the left ventricular outflow tract (A). The continuous-wave Doppler examination recorded from the left apical 5-chamber view confirms a markedly increased peak systolic aortic flow velocity with a high peak systolic trans-stenotic pressure gradient (ΔP = 149 mmHg). *Ao: aorta. LA: left atrium. LV: left ventricle.*

Each review allowed the official reviewer to categorize screened dogs regarding the AS and PS status, as "AS0" (absence of AS), "AS1", "AS2", or "AS3", and "PS0" (absence of PS), "PS1", "PS2", or "PS3", respectively corresponding to mild, moderate, and severe AS and PS, as defined above using ΔP values. The presence of other cardiac abnormalities (e.g., CHD, arrhythmias) was also recorded.

## Statistical analysis

For each screened Newfoundland dog, data of interest, i.e., main epidemiological features (age, body weight, sex), clinical (auscultation abnormalities), ECG findings, and Doppler variables (peak flow velocities and corresponding ΔP) were reported by the official reviewer in a spreadsheet (one per year). All spreadsheets (n = 14) were then gathered in one single file for the present study.

All statistics were performed by using a computer software (Addinsoft XLSTAT: statistical and data analysis solution, version 2020). Quantitative data were reported as median (interquartile range (IQR); minimal and maximal values) and percentage (%) when relevant. Quantitative values were analyzed using Mann-Whitney tests. Qualitative values were compared using a $Chi^2$ test or a Fisher exact test when the conditions for carrying out the $Chi^2$ test were not met. For all statistical analyses, the level of significance was set at $P < 0.05$.

## Results

### Epidemiological features of the study population

From January 2010 to December 2023, a total of 921 Newfoundland dogs (female:male sex ratio of 1.94; median age [IQR; minimum-maximum] = 1.9 years [1.6–2.7; 1.2–7.9] and body weight = 55.0 kg [50–60; 33.4–89]) underwent the cardiovascular screening program and were therefore included in the study. Females had a significantly lower body weight than males (55 kg [50–60; 33.4–89] *versus* 62 kg [56–66; 40–89], respectively; P < 0.0001).

### Aortic and pulmonic stenosis status

Cardiac status of the whole study population regarding AS and PS (i.e., respectively AS0 to AS3, or ASNC, and PS0 to PS3 or PSNC) is presented in Table 1. Doppler examinations could not allow the official reviewer to definitively conclude on AS and PS status for 61 dogs (6.6%) because the Doppler quality criteria (i.e., 2D images of good quality, clearly defined Doppler curves, correct placement of the Doppler cursor, or presence of concomitant ECG tracing) were not fulfilled. For these dogs, the reviewer asked for new Doppler examinations. However, these requested re-examinations were not performed, avoiding a definitive cardiac status conclusion. Therefore, a total of 860 Newfoundland dogs were definitively diagnosed for both AS and PS, 876 dogs for AS (i.e., 577 (65.9%) females and 299 (34.1%) males) and 879 dogs for PS (i.e., 299 (34.0%) males and 580 (66.0%) females). Out of the 860 screened Newfoundland dogs with a definitive AS and PS status, 851/860 (99.0%) were free of both AS and PS, with 868/876 (99.1%) and 877/879 (99.8%) being free of AS and PS, respectively. A total of 9/860 dogs (1.0%) were diagnosed with AS and/or PS: 8/876 (0.9%) were affected by AS (3 males and 5

**Table 1. Distribution of the 921 Newfoundland dogs that underwent the French cardiovascular breed screening program (2010–2023), according to their cardiac status regarding aortic stenosis (AS) and pulmonic stenosis (PS).**

| Cardiac status | Sub-classes | | | | | |
|---|---|---|---|---|---|---|
| **Aortic stenosis** | **AS0** | **AS1** | **AS2** | **AS3** | **ASNC** | **Total** |
| *Number of dogs* | 868 | 8 | 0 | 0 | 45 | 921 |
| *Percentage (%)* | 94.2 | 0.9 | 0 | 0 | 4.9 | 100 |
| **Pulmonic stenosis** | **PS0** | **PS1** | **PS2** | **PS3** | **PSNC** | **Total** |
| *Number of dogs* | 877 | 2 | 0 | 0 | 42 | 921 |
| *Percentage (%)* | 95.2 | 0.2 | 0 | 0 | 4.6 | 100 |

AS0, AS1, AS2, AS3: dogs with no AS, mild, moderate, and severe AS, respectively.

PS0, PS1, PS2, PS3: dogs with no PS, mild, moderate, and severe PS, respectively.

ASNC and PSNC: dogs with nonconclusive Doppler examinations of aortic and pulmonary flows, respectively.

females) and 2/879 (0.2%) by PS (2 females), with one female dog suffering from combined AS and PS.

Severity of AS and PS assessed by ΔP (mmHg) is shown in Table 2. No Newfoundland dogs were diagnosed neither with category AS2 or AS3 nor with category PS2 or PS3.

No significant sex predisposition was found for both AS and PS (P = 1.0 and P = 0.55, respectively). No significant difference regarding age of presentation was observed between AS-free dogs and dogs with AS (P = 0.45), and between PS-free dogs and dogs with PS (P = 0.30). No significant difference regarding body weight was observed between AS-free dogs and dogs with AS (P = 0.46), and between PS-free dogs and dogs with PS (P = 0.38).

Distribution of stenosis lesions (i.e., valvular, subvalvular, or supravalvular) was available for 4/8 Newfoundland dogs with AS. A single obstructive lesion was detected for these 4 dogs, and the subvalvular location was the most common AS form (n = 3/4). No dog was diagnosed with supra-valvular AS while a valvular obstruction was detected in 1/4 dog. The location of the stenosis in the case of PS was not specified for the two affected dogs.

## Cardiac abnormalities other than congenital VOTO

Out of the 921 screened Newfoundland dogs, 33 (3.6%) were found affected by at least one CHD other than AS and PS, i.e., mild aortic valve regurgitation (28/921; 3.0%) owing to quadricuspid aortic valve in one (1/921; 0.1%), mild mitral valve dysplasia (7/921; 0.8%), mild tricuspid valve dysplasia (2/921; 0.2%), and patent ductus arteriosus successfully treated using an

**Table 2. Peak systolic velocity (Vmax, m/s) and Doppler-derived pressure gradient (ΔP, mmHg) in Newfoundland dogs for which the board-certified specialist (VC) in charge of the French cardiovascular breed screening program (2010–2023) could conclude to a definitive cardiac status regarding aortic stenosis (n = 876) and pulmonic stenosis (n = 879) out of the 921 Newfoundland dogs that underwent the French cardiovascular breed screening program (2010–2023).** Dogs with other heart diseases (n = 33) were excluded from the analysis.

| Dog's category | Number of dogs | Median ΔP (mmHg) (interquartile range; minimum-maximum) | Median Vmax (m/s) (interquartile range; minimum-maximum) |
|---|---|---|---|
| **AS0** | 835 | 12 (10–15; 4–25) | 1.75 (1.57–1.94; 1–2.49) |
| **AS1** | 8 | 37 (29–41; 25–45) | 3.06 (2.67–3.21; 2.51–3.37) |
| **PS0** | 844 | 6 (5–8; 1–16) | 1.22 (1.08–1.39; 0.5–2) |
| **PS1** | 2 | 45 (43–47; 41–49) | 3.4 (3.3–3.4; 3.2–3.5) |

AS0, AS1: dogs with no AS and mild AS, respectively.

PS0, PS1: dogs with no PS and mild, respectively.

extra luminal ligation after intercostal thoracotomy (1/921; 0.1%). No electrocardiographic abnormalities were reported in any of the 921 tested dogs.

## Operators involved in the screening program

A total of 133 veterinarians were involved in the cardiovascular Newfoundland screening program, representing a total of 1014 examinations on the 921 dogs, 874 of which (94.9%) were examined in standing position, 47 (5.1%) in lateral recumbency. Out these 133 veterinarians, 11 (8.3%) were board-certified specialists in cardiology, while the remaining 122 (91.7%) were general practitioners.

Out of 921 tested Newfoundland dogs, nearly all (i.e., 915/921, 99.3%) were performed in metropolitan France (Fig 2), either in veterinary centers (n = 857/915, 93.7%) or by ambulatory veterinarians (n = 58/915, 6.3%).

Among the 876 and 879 dogs with a definitive cardiac status for AS and PS respectively, numbers of screening examinations needed to definitely conclude the cardiac status are presented in Table 3. For most dogs (i.e., 90.6% for AS and 91.0% for PS), the above-mentioned Doppler quality criteria were perfectly fulfilled allowing to establish a definitive cardiac status with one single imaging examination. For the remaining dogs with non-conclusive Doppler examinations, the latter were asked to be redone. The median (IQR; minimum-maximum) number of examinations required to conclude to the definitive cardiac status was 1 (1–1; 1–3) for AS and also for PS.

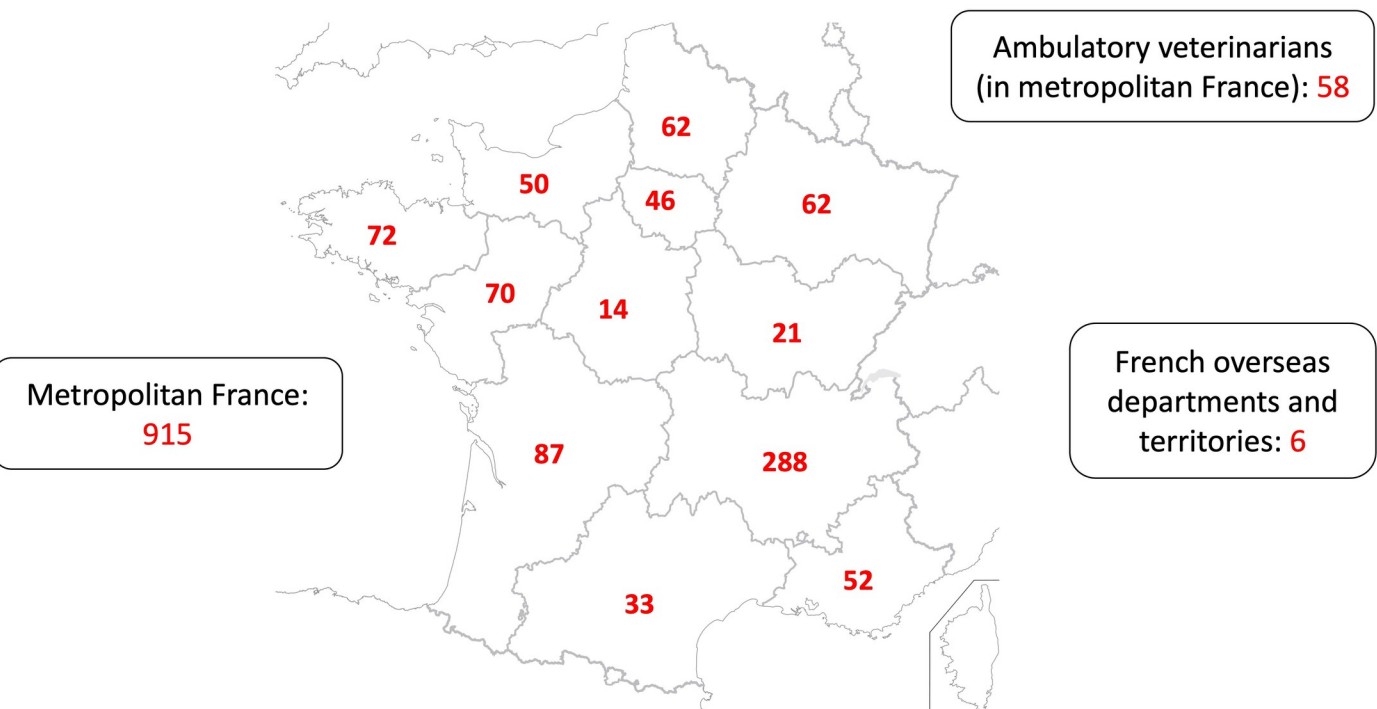

**Fig 2. Geographical distribution of echocardiographic examinations performed with 133 veterinarians on 921 Newfoundland dogs.** A total of 915 dogs were screened in metropolitan France, i.e., 857 in 61 different departments (corresponding to 12 regions) and 58 by ambulatory veterinarians. *Map of France from the Institut National de l'Information Géographique et Forestière, France (https://ign.fr/institut/ressources-pedagogiques –IGN 2016 –Open license).*

**Table 3. Number of Doppler examinations (i.e., 1, 2, or 3) that allowed the board-certified specialist (VC) in charge of the French cardiovascular breed screening program (2010–2023) to conclude to a definitive cardiac status regarding aortic stenosis (AS) and pulmonic stenosis (PS) (n = 876 for AS and n = 879 for PS out of the 921 Newfoundland dogs that underwent the screening program).**

| Stenosis status | Aortic stenosis category (number of dogs) | | Total of dogs | Percentage (%) |
|---|---|---|---|---|
| | AS0 | AS1 | | |
| **Number of examinations required to conclude** | | | | |
| 1 | 788 | 6 | 794 | 90.6 |
| 2 | 72 | 2 | 74 | 8.5 |
| 3 | 8 | 0 | 8 | 0.9 |
| **Total** | 868 | 8 | 876 | 100 |
| Stenosis status | *Pulmonic stenosis category (number of dogs)* | | Total of dogs | Percentage (%) |
| | PS0 | PS1 | | |
| **Number of examinations required to conclude** | | | | |
| 1 | 798 | 2 | 800 | 91.0 |
| 2 | 71 | 0 | 71 | 8.1 |
| 3 | 8 | 0 | 8 | 0.9 |
| **Total** | 877 | 2 | 879 | 100 |

AS0, AS1: dogs with no AS and mild AS, respectively.

PS0, PS1: dogs with no PS and mild PS, respectively.

All dogs (n = 92/921, 10.0%) screened by board-certified specialists (n = 11) were definitively diagnosed with AS and PS, with one single examination sufficient to reach a definitive diagnosis for all dogs regarding PS (92/92, 100%) and for 91/92 (98.9%) regarding AS. Conversely, regarding the 829/921 (90.0%) dogs screened by general practitioners, 45/829 (5.4%) and 42/829 (5.1%) dogs did not have a definitive diagnosis for AS and PS, respectively.

The median number of examinations required for board-certified specialists to classify dogs for AS and PS was significantly lower than for general practitioners (P = 0.04 and P = 0.005, respectively).

## Cardiac auscultation

Information concerning the presence or absence of a heart murmur was available for all the 921 Newfoundland dogs involved in the screening protocol. A left basilar systolic heart murmur was detected in the 9 dogs with AS and PS, and also in 3 out of the 818 dogs (0.4%) free of both AS and PS, and also free of any other heart disease. For these dogs, the heart murmur grade was low (1/6 for all dogs). No significant difference was observed between the number of left basilar systolic heart murmurs detected by board-certified specialists and by general practitioners (P = 0.09).

## Link between ventricular outflow tract obstructions and other factors

The inbreeding coefficient which was assessed from the pedigree analysis was available for all screened Newfoundland dogs with a definitive AS and PS status except one from Canadian origin (n = 859). No significant difference in inbreeding coefficients was found between dogs free of VOTO (median value = 0.39% [0–1.6; 0–25.8]) and those with AS (0.59% [0–0.8; 0–1.6]) or PS (0.39% [0.2–0.6; 0–0.8]; P = 0.86 and P = 0.72, respectively). Information regarding hips and elbow dysplasia was available from the French Kennel club for respectively 859 and 826 Newfoundland dogs. Hip dysplasia was not significantly associated with the diagnosis

either of AS or PS (respectively, P = 1 and P = 0.51). Similar results were observed regarding elbow dysplasia and both AS and PS (P = 1 for both categories).

## Proportion of dogs with AS and PS: Comparison between the start and the end of the breeding program

Data from the 14-year Newfoundland cardiovascular screening program (2010–2023) showed that no AS and PS were detected during the last 3 years of the program, the last AS and PS diagnosis being in 2019 and 2020, respectively.

A French referral cardiology center (Alfort Cardiology Unit, France) was chosen for evaluating canine breeds affected by AS and PS over time. During the study period (2010 to 2023), a total of 284 dogs of 58 different breeds were diagnosed in this center with either AS (n = 71, 23 breeds) or PS (n = 215, 51 breeds, including 2 dogs with both AS and PS). No Newfoundland dog was diagnosed with PS. Regarding AS, the five most commonly affected breeds were Boxer (n = 14; 19.7%), Bull Terrier (n = 10; 14.1%), Golden Retriever (n = 7 dogs; 9.9%), Newfoundland (n = 6 dogs; 8.5%), and White Swiss Shepherd (n = 5 dogs; 7.0%). During the last year of the program, no AS was diagnosed in Newfoundland dogs.

## Discussion

Newfoundland dogs are known to be predisposed to various heart diseases, i.e., dilated cardiomyopathy and also several CHD including patent ductus arteriosus and VOTO, more specifically SAS [1, 2, 8–12, 21]. In one study performed on a large veterinary hospital population (Veterinary Medical Teaching Hospital at California-Davis University, USA), Newfoundland dogs together with Bullmastiffs had the greatest prevalence (4.46%) and odds ratio (34.73) for SAS [12]. In another study including 390 dogs with AS from the same veterinary hospital, Newfoundland (6.80%) belonged to the top 5 affected pure breeds together with Boxer (4.49%), Bull Terrier (4.10%), Irish Terrier (3.13%), and Bouvier des Flandres (2.38%) [9]. Similarly, in one study performed in Europe (Italy), PS and SAS were the two most common diagnosed canine CHD (604/1132 cases, 53.4%), with large breed dogs including Newfoundland dogs being the most SAS affected canine breeds (odds ratios of 11.2, 9.4, and 7.0, for the Dogue de Bordeaux, Boxer, and Newfoundland breeds, respectively [1]). This high predisposition of Newfoundland dogs to heart diseases, and more specifically to congenital VOTO, has led to set up a national cardiovascular screening program in France in collaboration with the French Newfoundland kennel club, using both physical examination and echocardiography. Its main objective was to manage mattings with the long-term goal to reduce heart disease prevalence (and more specifically that of SAS) in the Newfoundland breed in this country. The present study provides an overview of the first 14 years of this cardiovascular screening program (2010–2023) with exhaustive analysis of the epidemiological, clinical, and echocardiographic data obtained from more than 900 tested dogs representing, to the best of our knowledge, one of the largest reported Newfoundland canine populations in veterinary cardiology.

Like the national cardiovascular screening program set up in France for the Boxer breed [17], the present screening program was mandatory since 2010 for any Newfoundland dog intended for breeding. Therefore, our study sample is more representative of the Newfoundland breeding population in France than the whole French Newfoundland population. This explains that the present study population includes nearly twice females than males (female: male sex ratio of 1.94), with a majority of young adult animals (75% of dogs were under than 2.7 years old with a median age of 1.9 years only).

One of the main features of this 14-year cardiovascular program is the large number of screened Newfoundland dogs, i.e., 921 dogs, with no missing clinical and imaging data for any of them, which represents one of the key strengths of the present study. This was achieved thanks to a strong involvement of veterinarians all over the country, either in veterinary centers or by ambulatory veterinarians, with a total of 133 operators involved in the program and 1014 examinations performed on the 921 screened dogs. Interestingly, as for the national cardiovascular screening program set up in France for the Boxer breed [17], very few operators (less than 10%) were board-certified specialists in veterinary cardiology, the large majority (91.7%) being non-specialist general practitioners. As the number of board-certified specialists was rather limited in France during the study period (11 veterinarians only), the only option to screen an optimal number of Newfoundland dogs all over the country was to involve non-specialist practitioners in the screening program, with supervision of all examinations by one single official board-certified specialist in order to ensure data quality. Although the large majority of Newfoundland dogs involved in the program were thus screened by non-specialists in veterinary cardiology (829/921, 90.0%), the official reviewer could definitively conclude on AS and PS status for most tested dogs (95.1% for AS and 95.4% for PS), and for most of them (i.e., 90.6% and 91.0%, respectively), a single examination allowed to obtain a definitive cardiac status, with less than 1% of dogs requiring 3 examinations and no dog requiring more than 3. These data are similar to those obtained with the national cardiovascular screening program set up for the Boxer breed [17]. They confirm results from other reports showing both the feasibility and advantages of involving non-cardiologist veterinarians with appropriate training in small animal cardiology for standard clinical and echocardiographic examinations [17, 22–24].

Nevertheless, in the present study, the number of examinations required for board-certified specialists to classify dogs for AS and PS was significantly lower than for general practitioners, with all dogs screened by the former being definitively diagnosed with AS and PS, whereas 5% of dogs screened by general practitioners could not have a definitive status for both AS and PS. This confirms that echocardiography is an observer's dependent imaging tool and that, from a practical point of view, the observer's experience has an impact on ability to diagnose canine heart diseases [25].

As expected, in the present study a left basilar systolic heart murmur was detected in all dogs with AS and PS. A similar but low-grade heart murmur (grade 1/6) was also detected in some apparently healthy Newfoundland dogs (less than 1%) as was shown with the national cardiovascular screening program set up in the Boxer breed (7.4% [17]). The prevalence of such innocent heart murmurs may have been underestimated owing to the typical thick double coat and large chest of Newfoundland dogs (making the detection of soft heart murmurs more challenging than in Boxer dogs), and also because of the high proportion of general practitioners involved in the program as compared to board-certified specialists, although the ability to detect a left basilar systolic heart murmur was not significantly different between the two. Cardiac auscultation ability has been demonstrated to depend on the level of the observer's experience [26, 27], and the very low number of dogs with a left basilar systolic heart murmur in the present study may have biased the present auscultatory findings. Nevertheless, the existence of soft innocent heart murmurs in the Newfoundland breed illustrates the need of echocardiographic examinations to differentiate nonpathological heart murmurs from those caused by mild VOTO (AS and PS), thus confirming the practical interest of the present cardiovascular screening program.

Despite the high reported predisposition of the Newfoundland breed to CHD, and more specifically SAS [1, 2, 8–12], the very large majority of the tested Newfoundland dogs with a definitive cardiac status (99.0%) were free of congenital VOTO. These dogs were therefore officially allowed for breeding, which may have contributed to improve the global breed's cardiac

health, as no AS and PS were detected during the last 3 years of the program (4 years for AS and 3 for PS). Additionally, less than 4% of all tested dogs were diagnosed with at least one CHD other than AS and PS, and as for all AS and PS cases, all these CHD were considered as mild except for one patient with patent ductus arteriosus that was further successfully surgically treated. Similarly, in the French referral cardiology center, Newfoundland dogs represented less than 10% of all dogs with AS, with no Newfoundland dog diagnosed with AS the last year of the program and none diagnosed with PS during the 14-year study period. Unlike reports from other European and North American countries [1, 2, 9, 12], these results suggest a relatively low prevalence of CHD including VOTO in the Newfoundland breed in France as well as the decreasing prevalence of the latter over time. This may be at least in part respectively related to the implemented cardiovascular screening program and to the low inbreeding level as confirmed by inbreeding coefficients, assessed from the pedigree analysis of all screened dogs except one from Canadian origin: inbreeding coefficients were not significantly different between dogs with either AS or PS and those without any arterial stenosis and were also very low (median values < 1% for each dog category, i.e., 0.59%, 0.39%, and 0.39%, respectively). These data contrast with results from another report including Newfoundland dogs from both Europe and North America. In this report, the Newfoundland population was highly inbred (average inbreeding coefficient of 6.99%), with SAS-affected dogs more often inbred and more closely related to each other than dogs free of SAS [10].

This study presents several limitations. The official board-certified specialist in charge of the program could not confirm data from physical examination and, owing to the large number of general practitioners involved in the screening and the potential difficulty of cardiac auscultation for some of these thick coat large chest dogs, the proportion of Newfoundland dogs with soft heart murmurs may have been underestimated. Additionally, as for the national Boxer cardiovascular screening program [17], left and right ventricular free wall thicknesses were used by the official board-certified specialist to validate echocardiographic and Doppler examinations. However, these ultrasound variables and the corresponding M-mode images were not included in the data base, and therefore were not available for the present retrospective study. Lastly, the board-certified official reviewer could not conclude to a definitive cardiac status for all tested dogs owing to inadequate quality of Doppler examinations. However, this represented 4.9% and 4.6% only of all screened Newfoundland dogs for AS and PS, respectively.

In conclusion, this 14-year national cardiovascular screening program set up in the Newfoundland breed has experienced a strong and successful involvement of veterinarians, breeders, and owners throughout France with no AS and PS detected during the last 3 years of the program, no AS and no PS detected neither in the French referral cardiology center respectively during the last year of the program and during the whole study period. This suggests the low and decreasing prevalence over time of congenital VOTO in the Newfoundland breed in this country. This illustrates also the need to maintain such a screening program over the next few years, in order to ensure as much as possible good cardiac health for Newfoundland breeding dogs and their offspring despite the known high predisposition of this canine breed to SAS and other heart diseases.

## Supporting information

**S1 File.**
(XLSX)

## Acknowledgments

We sincerely thank the "Foundation Un Cœur" under the aegis of the "Foundation de France" for supporting our scientific programs.

## Author Contributions

**Conceptualization:** Valérie Chetboul.

**Data curation:** Valérie Chetboul, Constance Fauveau.

**Formal analysis:** Constance Fauveau.

**Investigation:** Valérie Chetboul, Constance Fauveau, Peggy Passavin.

**Methodology:** Valérie Chetboul, Peggy Passavin.

**Project administration:** Valérie Chetboul.

**Software:** Constance Fauveau.

**Supervision:** Valérie Chetboul.

**Validation:** Valérie Chetboul.

**Visualization:** Valérie Chetboul.

**Writing – original draft:** Constance Fauveau.

**Writing – review & editing:** Valérie Chetboul, Constance Fauveau, Peggy Passavin.

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
