## [Decision Letter · Decision Letter 0]

4 Dec 2024

PONE-D-24-41219Congenital pulmonic and aortic stenosis in Newfoundland dogs: results of a 14-year French cardiovascular screening program (921 dogs)PLOS ONE

Dear Dr. Chetboul,

Thank you for submitting your manuscript to PLOS ONE. After careful consideration, we feel that it has merit but does not fully meet PLOS ONE’s publication criteria as it currently stands. Therefore, we invite you to submit a revised version of the manuscript that addresses the points raised during the review process.

We look forward to receiving your revised manuscript.

Kind regards,

Eyüp Serhat Çalık

Academic Editor

PLOS ONE

https://doi.org/10.1371/journal.pone.0285458

10.1371/journal.pone.0285458

In your revision ensure you cite all your sources (including your own works), and quote or rephrase any duplicated text outside the methods section. Further consideration is dependent on these concerns being addressed.

3. We note that your Data Availability Statement is currently as follows: [All relevant data are within the manuscript and its Supporting Information files]

4. We note that Figure 2 in your submission contain [map/satellite] images which may be copyrighted. All PLOS content is published under the Creative Commons Attribution License (CC BY 4.0), which means that the manuscript, images, and Supporting Information files will be freely available online, and any third party is permitted to access, download, copy, distribute, and use these materials in any way, even commercially, with proper attribution. For these reasons, we cannot publish previously copyrighted maps or satellite images created using proprietary data, such as Google software (Google Maps, Street View, and Earth). For more information, see our copyright guidelines: http://journals.plos.org/plosone/s/licenses-and-copyright.5*

1. You may seek permission from the original copyright holder of Figure 2 to publish the content specifically under the CC BY 4.0 license. 

Additional Editor Comments:

First of all, I wish the authors success in their work. It is a pleasing finding that the incidence of AS and PS in the Newfoundland dog breed is low and decreasing. Your manuscript was evaluated by two reviewers, and their suggestions are below. We look forward to your point-by-point responses to the reviewers' questions and to your resubmission of the manuscript after the revisions are made. Good luck.

Reviewers' comments:

Reviewer's Responses to Questions

**Comments to the Author**

1. Is the manuscript technically sound, and do the data support the conclusions?

Reviewer #1: No

Reviewer #2: Yes

2. Has the statistical analysis been performed appropriately and rigorously? 

Reviewer #1: No

Reviewer #2: Yes

3. Have the authors made all data underlying the findings in their manuscript fully available?

Reviewer #1: Yes

Reviewer #2: Yes

4. Is the manuscript presented in an intelligible fashion and written in standard English?

Reviewer #1: Yes

Reviewer #2: Yes

5. Review Comments to the Author

Reviewer #1: 1. Line 66: The statement "high relative risk for Newfoundland dogs to develop subvalvular AS" is vague and lacks specificity. Given that you have referenced eight studies, it would be beneficial to quantify the relative risk by providing an average percentage or a range of prevalence rates from the referenced studies. This would lend greater clarity and support to your point of view.

2. Prevalence of AS and PS: In the results section (lines 194-202 and Table 1), it appears that only 8 dogs were diagnosed with AS, 2 with PS, and 1 with both AS and PS. This corresponds to a low prevalence of 0.9% for AS and 0.2% for PS in the screened population. Additionally, all diagnoses fall within the mild category (lines 203-204, lines 159-161). This low prevalence contradicts findings from the existing literature, which report a higher percentage, as noted in your introduction (lines 64-67). Please reconcile these discrepancies and clarify the expected prevalence of these conditions in Newfoundland dogs.

3. Statistical Validity of Coat Color Analysis: You mention in lines 43-44 and at line 316-320 that "no link was made between the diagnosis of AS and PS and coat color (black, black and white, and brown)." Given that your disease group includes only 11 animals (8 with AS, 2 with PS, and 1 with both AS and PS), and the coat color analysis includes 3 color categories, , the sample size is too small to draw statistically valid conclusions about the relationship between coat color and these conditions. From a statistical perspective, this analysis is unlikely to yield meaningful results. Please reconsider this analysis or provide further justification for your findings.

4. Supra-Valvular AS (SAS) Diagnosis: In line 214, you state that "no dog was diagnosed with supra-valvular AS (SAS)." However, the introduction of your paper places significant emphasis on SAS as a key condition of interest.

5. Plagiarism Concern: The paper exhibits a high degree of similarity in the M&M section with the article published at https://doi.org/10.1371/journal.pone.0285458. I would recommend addressing this issue of potential overlap and ensuring that all references are appropriately cited to avoid concerns of plagiarism.

Reviewer #2: 1. My suggestions is to incorporate the physiology of pulmonary and aortic stenosis in discussion section.

2. line 189 to line 92 is not clear, rewrite

3. line263 to line 265 not clear

4. Line 302, make it easy

5. line 334-335 not clear

6. PLOS authors have the option to publish the peer review history of their article (what does this mean?). If published, this will include your full peer review and any attached files.

Reviewer #1: No

Reviewer #2: **Yes: **Ijaz Ahmad

---

## [Author Response · Author response to Decision Letter 0]

19 Dec 2024

ASSOCIATE EDITOR

Comments to the Author:

First of all, I wish the authors success in their work. It is a pleasing finding that the incidence of AS and PS in the Newfoundland dog breed is low and decreasing. Your manuscript was evaluated by two reviewers, and their suggestions are below. We look forward to your point-by-point responses to the reviewers' questions and to your resubmission of the manuscript after the revisions are made. Good luck.

The authors would like to sincerely thank the Associate Editor for his/her help in improving the manuscript and for his/her encouragements. We are very pleased that experts in the field were able to review our paper. We have taken into account all the editor and reviewers’ comments, which have improved the quality of our article. You will find below our point-by-point responses to the reviewers' questions. 

REVIEWER 1

- Comment #1: Line 66: The statement "high relative risk for Newfoundland dogs to develop subvalvular AS" is vague and lacks specificity. Given that you have referenced eight studies, it would be beneficial to quantify the relative risk by providing an average percentage or a range of prevalence rates from the referenced studies. This would lend greater clarity and support to your point of view.?

We thank the reviewer for his/her comment. However, the most relevant prevalence rates and odds ratio values are provided and discussed on page 19 (lines 351-361) of the new version of the manuscript : “In one study performed on a large veterinary hospital population (Veterinary Medical Teaching Hospital at California-Davis University, USA), Newfoundland dogs together with Bullmastiffs had the greatest prevalence (4.46%) and odds ratio (34.73) for SAS [12]. In another study including 390 dogs with AS from the same veterinary hospital, Newfoundland (6.80%) belonged to the top 5 affected pure breeds together with Boxer (4.49%), Bull Terrier (4.10%), Irish Terrier (3.13%), and Bouvier des Flandres (2.38%) [9]. Similarly, in one study performed in Europe (Italy), PS and SAS were the two most common diagnosed canine CHD (604/1132 cases, 53.4 %), with large breed dogs including Newfoundland dogs being the most SAS affected canine breeds (odds ratios of 11.2, 9.4, and 7.0, for the Dogue de Bordeaux, Boxer, and Newfoundland breeds, respectively [1]).” However, in order to satisfy the reviewer’s comment, the corresponding sentence has been completed as follows on page 4 (lines 66-67) of the new version of the manuscript: “Aortic stenosis (AS) and pulmonic stenosis (PS) are two of the most common congenital heart diseases (CHD) in dogs both consisting in respectively left and right ventricular outflow tract obstructions (VOTO) [1-7], with a high relative risk for Newfoundland dogs to develop subvalvular AS (SAS), odds ratios reaching up to 35 as compared with other canine breeds [1,2,8-13].” 

- Comment #2: Prevalence of AS and PS: In the results section (lines 194-202 and Table 1), it appears that only 8 dogs were diagnosed with AS, 2 with PS, and 1 with both AS and PS. This corresponds to a low prevalence of 0.9% for AS and 0.2% for PS in the screened population. Additionally, all diagnoses fall within the mild category (lines 203-204, lines 159-161). This low prevalence contradicts findings from the existing literature, which report a higher percentage, as noted in your introduction (lines 64-67). Please reconcile these discrepancies and clarify the expected prevalence of these conditions in Newfoundland dogs.

We agree with the reviewer about discrepancies between the low and decreasing prevalence of PS and AS in the Newfoundland breed in France and the existing literature from other countries. This interesting point is discussed on pages 22-23 (lines 428-452) as follows: “Despite the high reported predisposition of the Newfoundland breed to CHD, and more specifically SAS [1,2,8-12], the very large majority of the tested Newfoundland dogs with a definitive cardiac status (99.0%) were free of congenital VOTO. These dogs were therefore officially allowed for breeding, which may have contributed to improve the global breed’s cardiac health, as no AS and PS were detected during the last 3 years of the program (4 years for AS and 3 for PS). Additionally, less than 4% of all tested dogs were diagnosed with at least one CHD other than AS and PS, and as for all AS and PS cases, all these CHD were considered as mild except for one patient with patent ductus arteriosus that was further successfully surgically treated. Similarly, in the French referral cardiology center, Newfoundland dogs represented less than 10% of all dogs with AS, with no Newfoundland dog diagnosed with AS the last year of the program and none diagnosed with PS during the 14-year study period. Unlike reports from other European and North American countries [1,2,9,12], these results suggest a relatively low prevalence of CHD including VOTO in the Newfoundland breed in France as well as the decreasing prevalence of the latter over time. This may be at least in part respectively related to the implemented cardiovascular screening program and to the low inbreeding level as confirmed by inbreeding coefficients, assessed from the pedigree analysis of all screened dogs except one from Canadian origin: inbreeding coefficients were not significantly different between dogs with either AS or PS and those without any arterial stenosis and were also very low (median values < 1% for each dog category, i.e., 0.59%, 0.39%, and 0.39%, respectively). These data contrast with results from another report including Newfoundland dogs from both Europe and North America. In this report, the Newfoundland population was highly inbred (average inbreeding coefficient of 6.99%), with SAS-affected dogs more often inbred and more closely related to each other than dogs free of SAS [10]. ”

- Comment #3: Statistical Validity of Coat Color Analysis: You mention in lines 43-44 and at line 316-320 that "no link was made between the diagnosis of AS and PS and coat color (black, black and white, and brown)." Given that your disease group includes only 11 animals (8 with AS, 2 with PS, and 1 with both AS and PS), and the coat color analysis includes 3 color categories, , the sample size is too small to draw statistically valid conclusions about the relationship between coat color and these conditions. From a statistical perspective, this analysis is unlikely to yield meaningful results. Please reconsider this analysis or provide further justification for your findings.

We agree with the reviewer’s comment. All data and statements regarding coat colors have been deleted from the new version of the manuscript. 

- Comment #4: Supra-Valvular AS (SAS) Diagnosis: In line 214, you state that "no dog was diagnosed with supra-valvular AS (SAS)." However, the introduction of your paper places significant emphasis on SAS as a key condition of interest.

We thank the reviewer for his/her comment. Nevertheless, SAS is the abbreviation for subvalvular stenosis (and not supravalvular stenosis). As described in the veterinary literature, in the present study, the subvalvular location was the most common AS form (i.e., SAS) while no dog was diagnosed with supravalvular AS (see page 11, lines 222-225).

- Comment #5: Plagiarism Concern: The paper exhibits a high degree of similarity in the M&M section with the article published at https://doi.org/10.1371/journal.pone.0285458. I would recommend addressing this issue of potential overlap and ensuring that all references are appropriately cited to avoid concerns of plagiarism.

We understand the reviewer’s comment, as the protocol designs of both studies (the present one and our previous publication (https://doi.org/10.1371/journal.pone.0285458) on the same topic but dedicated to another canine breed, entitled “Congenital ventricular outflow tract obstructions in Boxer dogs: Results of a 17-year cardiovascular breed screening program in France (3126 dogs)” are identical. 

Nevertheless, the accusation of plagiarism is a serious charge. Plagiarism is defined as the process or practice of using ANOTHER person's ideas or work and pretending that it is your own (https://dictionary.cambridge.org/dictionary/english/plagiarism). The article published at https://doi.org/10.1371/journal.pone.0285458 is from our group and for both studies, the national cardiovascular screening programs were implemented by the SAME first author of the two papers and are perfectly similar AS REQUESTED by the national French kennel club. The first screening program dedicated to the Boxer breed (described in the 2023 Plos One paper) started in 2005 and because it worked well for owners, breeders, and veterinarians, the second screening program dedicated to the Newfoundland breed was set up 5 years later and was perfectly similar to the former according to national French kennel club’s request. 

We had therefore two choices for the present paper: either to entirely refer to the first article or to, again, briefly describe the cardiovascular screening protocol to make the reading more understandable, particularly for novice readers. We have chosen the second option. However, to avoid any confusion for both the reviewer and the reader, the following sentences have been added to the new version of the manuscript on page 6 (lines 121-126) as follows: “The protocol was strictly similar to that of the national mandatory cardiovascular breed screening program set up in France for the Boxer breed [17]. The two screening programs (the present one on Newfoundland dogs and the previous one on Boxer dogs) were designed by the same board-certified cardiologist (VC) who was also the single official reviewer of Doppler examinations for both breeds, as requested by the national French kennel club [17].”

REVIEWER 2

Comment #1: My suggestions is to incorporate the physiology of pulmonary and aortic stenosis in discussion section. 

We thank the reviewer for his/her suggestion. Some pathophysiological features of both pulmonary and aortic stenoses were already included in the introduction section on page 4 (previous lines 64-75) : “Aortic stenosis (AS) and pulmonic stenosis (PS) are two of the most common congenital heart diseases (CHD) in dogs both consisting in respectively left and right ventricular outflow tract obstructions (VOTO) [1-7], with a high relative risk for Newfoundland dogs to develop subvalvular AS (SAS) [1,2,8-13]. Subvalvular AS is characterized by various types of lesions, including fibrous or fibromuscular nodules, ridges, rings, and complex tunnel-like tissues located below the aortic valve, thus creating discrete to severe obstruction to blood flow across the left ventricular outflow tract [8,13-15]. Dogs with moderate to severe SAS have been demonstrated to be at higher risk for developing clinical signs (e.g., exercise intolerance, syncope), sudden death, infective endocarditis, and left-sided congestive heart failure, with a median survival of 3.0 years only for dogs with Doppler-derived peak systolic trans-stenotic pressure gradient (�P) > 130 mmHg [14,15].” Nevertheless, in order to satisfy the reviewer’s request, the previous paragraph has been completed as follows on page 4 (current lines 67-71) of the new version of the manuscript: “In both AS and PS cases, obstruction to the ejection of blood from respectively the left and right ventricles leads to turbulences associated with increased of blood velocity across the stenotic lesion depending on the obstruction severity, and thus to the presence of a left basilar systolic heart murmur detected on cardiac auscultation”. 

Comment #2: line 189 to line 92 is not clear, rewrite

As requested by the reviewer, the corresponding paragraph has been reworded as follows on page 10 (lines 197-202) of the new version of the manuscript: “Cardiac status of the whole study population regarding AS and PS (i.e., respectively AS0 to AS3, or ASNC, and PS0 to PS3 or PSNC) is presented in Table 1. Doppler examinations could not allow the official reviewer to definitively conclude on AS and PS status for 61 dogs (6.6%) because the above-mentioned Doppler quality criteria (i.e., 2D images of good quality, clearly defined Doppler curves, correct placement of the Doppler cursor, or presence of concomitant ECG tracing) were not fulfilled” (instead of “Cardiac status of the whole study population regarding AS and PS is presented in Table 1. Doppler examinations could not allow the official reviewer to definitively conclude on AS and PS status for 61 dogs (6.6%) because the above-mentioned Doppler quality criteria were not fulfilled.”). 

Comment #3: line263 to line 265 not clear

As requested by the reviewer, the corresponding paragraph has been reworded as follows on page 15 (lines 274-278): “For most dogs (i.e., 90.6% for AS and 91.0% for PS), the above-mentioned Doppler quality criteria were perfectly fulfilled allowing to establish a definitive cardiac status with one single imaging examination. For the remaining dogs with non-conclusive Doppler examinations, the latter were asked to be redone. The median (IQR; minimum-maximum) number of examinations required to conclude to the definitive cardiac status was 1 (1-1; 1-3) for AS and also for PS. ” (instead of “For most dogs (i.e., 90.6% for AS and 91.0% for PS), a single examination allowed to obtain a definitive cardiac status. The median (IQR; minimum-maximum) number of examinations required to conclude to the definitive cardiac status was 1 (1-1; 1-3) for AS and also for PS.”). 

Comment #4: Line 302, make it easy

As requested by the reviewer, the corresponding sentence has been reworded as follows on page 17 (lines 315-317): “No significant difference was observed between the number of left basilar systolic heart murmurs detected by board-certified specialists and by general practitioners (P = 0.09).” (instead of “The ability to detect a left basilar systolic heart murmur was not significantly different between board-certified specialists and general practitioners (P = 0.09).”). 

Comment #5: line 334-335 not clear

As requested by the reviewer, the corresponding sentence has been reworded as follows on page 18 (lines 341-345): “Regarding AS, the five most commonly affected breeds were Boxer (n = 14; 19.7%), Bull Terrier (n = 10; 14.1%), Golden Retriever (n = 7 dogs; 9.9%), Newfoundland (n = 6 dogs; 8.5%), and White Swiss Shepherd (n = 5 dogs; 7.0%). During the last year of the program, no AS was diagnosed in Newfoundland dogs.” (instead of “Regarding AS, the five most commonly affected breeds were Boxer (n = 14; 19.7%), Bull Terrier (n = 10; 14.1%), Golden Retriever (n = 7 dogs; 9.9%), Newfoundland (n = 6 dogs; 8.5%), and White Swiss Shepherd (n = 5 dogs; 7.0%), with no Newfoundland dog diagnosed with AS the last year of the program”.).

---

## [Decision Letter · Decision Letter 1]

14 Jan 2025

Congenital pulmonic and aortic stenosis in Newfoundland dogs: results of a 14-year French cardiovascular screening program (921 dogs)

PONE-D-24-41219R1

Dear Dr. Ghetboul,

We’re pleased to inform you that your manuscript has been judged scientifically suitable for publication and will be formally accepted for publication once it meets all outstanding technical requirements.

Kind regards,

Eyüp Serhat Çalık

Academic Editor

PLOS ONE

Additional Editor Comments (optional):

Reviewers' comments:

Reviewer's Responses to Questions

**Comments to the Author**

1. If the authors have adequately addressed your comments raised in a previous round of review and you feel that this manuscript is now acceptable for publication, you may indicate that here to bypass the “Comments to the Author” section, enter your conflict of interest statement in the “Confidential to Editor” section, and submit your "Accept" recommendation.

Reviewer #2: All comments have been addressed

Reviewer #3: All comments have been addressed

2. Is the manuscript technically sound, and do the data support the conclusions?

Reviewer #2: Yes

Reviewer #3: Yes

3. Has the statistical analysis been performed appropriately and rigorously? 

Reviewer #2: Yes

Reviewer #3: Yes

4. Have the authors made all data underlying the findings in their manuscript fully available?

Reviewer #2: Yes

Reviewer #3: Yes

5. Is the manuscript presented in an intelligible fashion and written in standard English?

Reviewer #2: Yes

Reviewer #3: Yes

6. Review Comments to the Author

Reviewer #2: thank you for incorporating the suggested changes into the manuscript. I have reviewed the revised version and am satisfied with the updates. the manuscript is now in a much-improved form, and i have no further comment at this stage.

Reviewer #3: (No Response)

7. PLOS authors have the option to publish the peer review history of their article (what does this mean?). If published, this will include your full peer review and any attached files.

Reviewer #2: **Yes: **Ijaz Ahmad

Reviewer #3: No

---

## [Editor Report · Acceptance letter]

22 Jan 2025

PONE-D-24-41219R1 

PLOS ONE

Dear Dr. Chetboul, 

I'm pleased to inform you that your manuscript has been deemed suitable for publication in PLOS ONE. Congratulations! Your manuscript is now being handed over to our production team.

Kind regards, 

on behalf of

Dr. Eyüp Serhat Çalık 

Academic Editor

PLOS ONE